# Risk Factors Related to Eating Disorders in a Romanian Children Population

**DOI:** 10.3390/nu15132831

**Published:** 2023-06-21

**Authors:** Bianca-Teodora Ciurez, Oana-Claudia Cobilinschi, Anamaria-Renata Luca, Iulia Florentina Țincu, Doina Anca Pleșca

**Affiliations:** 1“Dr. Victor Gomoiu” Clinical Children Hospital, 022102 Bucharest, Romania; biancaciurez@yahoo.com (B.-T.C.); anamaria.luca01@gmail.com (A.-R.L.); iulia.tincu@umfcd.ro (I.F.Ț.); doina.plesca@umfcd.ro (D.A.P.); 2Faculty of Medicine, “Carol Davila” University of Medicine and Pharmacy, 050474 Bucharest, Romania; 3“St. Marie” Clinical Hospital, 011172 Bucharest, Romania

**Keywords:** ARFID, eating disorders, food refusal, nutritional intervention, PARDI

## Abstract

(1) Background: The complex known as avoidant/restrictive food intake disorder (ARFID) is one of the eating disorders that cannot be explained by chronic disease. The aim of this study was to determine the characteristics of patients who were identified as being at risk of developing ARFID and referred to paediatricians, according to their age and duration of symptoms. (2) Methods: Children aged 2–10 years (Group 1) presenting with eating disorders were initially recruited in the “Dr. Victor Gomoiu” Clinical Children Hospital in Bucharest. Group 2 included patients presenting for routine paediatric visits as controls. The study population was given the PARDI questionnaire as well as questions related to demographics, screening growth and development, physical and mental background, and current feeding and eating patterns. Items were scored on a 7-point scale ranging from 0 to 6. (3) Results: A total of 98 individuals were divided equally into the two study groups. There was no difference in terms of sex, living area, mothers’ education level or living standards between the two groups. ARFID children were more likely to be underweight, were unsuccessful at weaning or have irregular feeding habits and a history of allergies. The mean age of onset for chronic symptoms was significantly lower than the onset of acute food refusal—4.24 ± 2.29 vs. 6.25 ± 3.65, *p* = 0.005. (4) Conclusions: feeding disorders are an important issue among paediatricians, and a proper awareness of them when treating these patients should be included in daily practice.

## 1. Introduction

The complex known as avoidant/restrictive food intake disorder (ARFID) is considered one of the eating disorders that cannot be explained by chronic disorders [1]. Children experiencing ARFID present with sensory aversions to various food types based on aspects of the foods, such as taste, texture and smell, and often report symptoms related to food intake (regurgitation, pain, nausea), eventually resulting in food refusal. As a common result of a inappropriate macro- and micronutrient intake, their nutritional status is affected, and sometimes patients present with decreased blood vitamin levels [2]. These disorders also have a social and behavioural impact on patients due to the challenges that can arise during collective eating or social and family gatherings that involve food and can lead to feelings of frustration and/or decreased self-esteem [3,4,5]. One can find ARFID in the DSM-5 and the International Classification of Diseases, Tenth Edition (ICD-10), where it was added in response to the need to characterise the eating behaviours of patients that can neither meet the criteria for anorexia nervosa or bulimia nervosa and that cannot be explained by an underlying disease or a developmental delay [6]. There is a lack of studies performed on epidemiological issues for a paediatric population, so there is not a clear image of the magnitude of the disease, and the age of incidence seems to be younger than comparable onset times [7,8]. The mechanisms of the disorders are yet to be fully understood, although some progress has been made in recent years in terms of plurifactorial involvement, such as parents’ attitude towards eating habits, eating experiences during the first years of life and children’s general susceptibility [9]. ARFID is also more often associated with a medical history involving prematurity, genetic disorders, and various gastrointestinal and neurological chronical conditions [10,11]. Both a child’s relationship with food and eating habits have a significant relationship with their temperament, neurodevelopmental level and intellectual perception of food [12,13,14]. In general, children with ARFID consume whatever they consider to be safe, sometimes very energy dense foods and beverages, so normal or even overweight patients can also be diagnosed with restricted food intake, although it makes difficult for clinicians to fulfil the criteria.

As a working tool, the Pica, ARFID, and Rumination Disorder Interview (PARDI) is a validated semistructured, multi-informant clinical assessment designed to assess and diagnose ARFID and other eating disorders according to the DSM-5 criteria [15]. The questionnaire provides severity scales for the conditions. There is a double burden of ARFID in relation to psychological development: coexistence with parents’, (mainly mothers’) depression, somatization, mental stress or past negative experience and a lack of autonomy and initiative as a result of the children’s relationship with food.

Aims of the study: The primary objective of the present study was to determine the characteristics of patients who were identified as being at risk of eating disorders and referred to a paediatric gastroenterologist. A secondary outcome was to determine whether children who met the ARFID criteria had different severity profiles according to their age and duration of symptoms.

## 2. Materials and Methods

### 2.1. Participants and Procedures

This was a prospective, case–control study comparing patients presenting with food refusal with normal, healthy controls sampled 1:1 and matched by age. It included children presenting with eating difficulties that were initially recruited during a two-year period (2020–2022), i.e., in the SARS-CoV-2 pandemic, from a large cohort of the paediatric population who were referred to the paediatric gastroenterology service at the “Dr. Victor Gomoiu” Clinical Children’s Hospital, Bucharest, Romania. The following are the inclusion criteria for study population: (1) aged 2–10 years; (2) addressed for evaluation of eating disorder symptoms; (3) informed consent signed by the caregiver. Individuals identified by an paediatric gastroenterologist as being at risk of ARFID, were considered as Group 1. In comparison, Group 2 included patients present in the waiting area of the same setting awaiting a regular paediatric visit who were randomly selected and had no disturbances in food intake. Patients with acute or chronic gastrointestinal disorders or with any other criteria for comorbid medical disorders known to influence eating or weight were excluded from both groups. For the secondary outcomes, patients at risk of ARFID were divided according to age, i.e., less than 5 years and between 5 and 10 years. Moreover, with regard to the duration of symptoms, the population included in the study was assessed in two different groups, i.e., duration of symptoms less than 12 months and over 12 months. Data collection was based on face-to-face questionnaires or telephone interviews performed by the same paediatric gastroenterologist. In the procedure, caregivers were assured of the voluntary nature of participation and confidentiality of the research and provided informed consent before entering the study protocol. The research project was approved by the Hospital Committee of Ethics, no. 1758/01.02.2022, that reviewed the study design and all informational material.

### 2.2. Measures

Demographic characteristics: Mothers of the children were asked to answer questions concerning their child’s date of birth, sex, type of gestational period (normal or pathologic), birth weight in grams, chronic somatic diseases, neurodevelopmental, and mental disorders and intellectual disability (as an exclusion criteria). Living area, parents age in years, parental education, economic status and marital status were also taken into consideration. Nutritional status was determined according to Z score for BMI, adjusted for age and sex.

The feeding description was evaluated as follows: type of nutrition in the first 6 months of life (breastfeeding = BE, formula fed = FF, mixt feeding = MF), successful weaning at 7 months of age (yes/no answers), regular type of food intake (mothers were offered explanations for consideration. Regular feeding was considered as three meals per day and two snacks, while irregular feeding was considered the pattern characterized by serving meals whenever the child wanted to eat). For this item, yes/no answers were accepted, as well as for any anterior or present documented diagnosis of food allergy (yes/no answers).

Nutritional intervention was also addressed in the protocol. Patients were divided and described in terms of oral nutrient supplementation, namely if they were prescribed special hypercaloric formulas, enteral feeding using nasogastric tube placement (specially maintained during hospitalization) or dietary indicators. Additionally, individuals receiving partial parenteral nutrition were analysed in the study. During the evolution for nutritional recovery, patients were registered for their duration of nutrition therapy in terms of days.

The PARDI questionnaire, as validated by Bryant-Waugh et al. [15], includes an introduction with items assessing growth, development, physical and/or mental health conditions, and current patterns of feeding and/or eating that would rule out a feeding disorder diagnosis. The following items are intended to inform the diagnostic algorithm, to provide severity ratings and to characterise in three profiles of severity for sensory sensitivity: lack of interest in eating and fear of aversive consequences. We applied the questions to screen for the outset, growth and development, physical and mental checklist criteria, current feeding and eating patterns, and then the ARFID diagnostic items. The majority of the items are scored on a 7-point scale ranging from 0 (no symptoms) to 6 (severe symptoms).

### 2.3. Statistical Analysis

Data regarding the demographic, medical, and anthropometric measures were calculated as means and standard deviations for continuous variables and proportions for the categorical variables. Differences among groups were compared using the chi-square test or fisher exact test for the categorical variables and the *t*-test for the continuous variables. Differences between groups were assessed with *p* < 0.05. All statistical analyses were performed in SPSS version 20.

## 3. Results

### 3.1. Baseline Characteristics

There was initially a total of 125 patients presented for feeding difficulties, out of which 101 (80.8%) were admitted to the study population after excluding individuals with chronic medical conditions. After the protocol explanation and informed consent, only 98 (78.4%) were finally admitted as the study population and are equally represented in the two study groups. The baseline characterization of the study individuals is shown in Table 1. There was no difference in terms of sex, living area, mothers’ educational level or living standards distribution in the two groups. The majority (79.59%) of caregivers reported a normal gestational period for their children with no significant differences between the groups (*p* = 0.316). The mother’s average age at the time of the interview was significantly higher in the study group as well as fathers’ mean age (29.43 ± 3.81 vs. 27.03 ± 3.04, *p* < 0.001 and 34.09 ± 5.77 vs. 31.88 ± 3.94, *p* = 0.004, respectively). Individuals at risk of ARFID had the lowest body mass index Z score when adjusted for age and sex, and this differs significantly between the two groups (Table 1).

As expected, there were significant group differences in terms of regular feeding patterns. Mothers reported difficulties in achieving successful weaning at the age of 7 months in Group 1, and the difference was statistically significant between the groups. Patients at risk of ARFID were mostly feed in the 6 months of life with formula or a combination of breastmilk and formula milk when compared to Group 2.

### 3.2. Age and Duration of Symptom Influencing Risk of ARFID

The mean age for chronic onset symptoms was significantly lower than the acute onset of food refusal, 4.24 ± 2.29 vs. 6.25 ± 3.65, *p* = 0.005 (Table 2). Males were significantly more likely to experience chronic rather than acute symptoms (*p* = 0.005), while female individuals were significantly more likely to experience the acute onset of feeding problems (*p* = 0.005). We found significant age group differences in the ARFID section of the PARDI questionnaire. Patients less than 5 years had higher mean scores for sensory profiles, lack of interest profile, concern profile and ARFID severity scale.

The same variation was met by patients having chronic onset symptoms. Individuals with chronic ARFID symptoms presented with significantly lower BMI Z scores compared to patients with ARFID symptoms developed for less than 12 months (chronic = −1.88; acute = −1.22; *p* = 0.05), and younger ages also had significantly lower BMI Z scores than patients over 5 years (−1.94 ± 0.33 vs. −1.10 ± 0.88, *p* = 0.05).

### 3.3. Eating Patterns

In the variables considered within the area of dysfunctional eating patterns, ‘number of meals’ (*p* = 0.001) and using special diets (*p* = 0.001) are related to the risk of eating disorders. In the meantime, there was no difference for the variables of eating ‘at least 3 meals a day’ (*p* = 0.077), as well as ‘having breakfast’ (*p* = 0.462). Practicing regular ‘exercise’ (*p* = 0.377), and ‘exercise frequency’ (*p* = 0.314) were not statistically significant variables for the risk of eating disorders (Table 3).

Subjects with confirmed allergy diagnoses have a higher risk of developing eating disorders. The risk of belonging to the risk group for eating disorders was augmented by 2.4 times if there was a food allergy diagnosis in the first year of life (OR = 2.4; 95% CI 0.85–0.95; *p*-value < 0.001).

### 3.4. Nutritional Intervention

After analysing therapeutic interventions from the point of view of oral, tube feeding, parenteral or stomal nutritional supplementation, in the target group, 65.3% (*n* = 32) oral, 28.5% (*n* = 14) enteral, 20.4% (*n* = 10) parenteral, and 6.1% (*n* = 3) stomal intervention was observed. In the control group, 57.1% (*n* = 28) oral, 10.2% (*n* = 5) enteral, and 6.1% (*n* = 3) parenteral nutrition were observed. Figure 1 schematically expresses this information. By applying the chi-square test, the following results were obtained: χ^2^(1.98) = 0.688, *p* = 0.407 for enteral nutrition; χ^2^(1.98) = 5.288, *p* = 0.021 for enteral nutrition; χ^2^(1.98) = 4.346, *p*= 0.037 for parenteral nutrition; and χ^2^(1.98) = 3.095, *p* = 0.079 for nutrition using a stoma. There may be a two-way statistical association between the risk of ARFID and the need for enteral and parenteral nutrition. Figure 1 shows the median duration of the nutrition intervention in the two selected patients groups according to interventional type.

## 4. Discussion

In the context of considering ARFID a debilitating complex disorder with unclear aetiology, some genetic determinants and important knowledge gaps, it is reasonable to make important methodological considerations for understanding the determinants and predispositions factors. The current study was aimed to reveal the characteristics of children at risk of ARFID when the evaluation was made by a paediatrician.

As we know so far, there is a male prevalence of ARFID diagnosis [16], but in our study population, there was a similar sex distribution when compared to nonfeeding disorders in children. There were some differences and similarities when compared to the standard population. Specifically, patients at risk of ARFID did not differ in terms of mean age, living area or standards, nor regarding pregnancy evolution or birth weight.

Patients with typical behaviours of ARFID were significantly more likely to have an irregular feeding pattern. The interview related that eating pattern is made through playing, colouring, watching cartoons and singing; nevertheless, such compartmental therapy during meals is considered to reduce anxiety, prepares the child to accept new foods and helps them feel safe about new sensory profiles. This can take up to several months until the patient is ready to accept new foods into their diet [17]. Such an intervention was created and is based upon the mechanistic hypothesis, and the authors developed an interoceptive and exteroceptive treatment model. Children were provided frameworks whereby they could perceive food sensations with curiosity but not with fear, in order to achieve food tolerance, and the focus is on describing the food’s texture and visual perception [17].

Younger children with ARFID were more likely to experience significant weight loss or failure to reach appropriate levels of growth. Moreover, they were more likely to be addressed by oral or enteral nutritional supplementation techniques, with significantly higher mean scores for fears of choking and/or vomiting, and texture and/or sensitivity issues regarding food. These aspects are similar with those in previous studies, first in early onset eating disorders research [18,19], then in a multicentre study focused mainly on ARFID [20].

We found that the younger the age is, the longer period of food refusal is reported by the parents. In our study, males were more likely to have higher rates of chronic onset symptoms than females. Our research was focused on several previous studies that evaluated the emotional eating behaviours or food sensitivity problems in children with feeding difficulties [21]. Children with chronic ARFID symptoms presented with significantly poorer nutritional status compared to those with acute symptoms. This finding is consistent with previous reports confirming that chronicity is typically associated with more negative outcomes in eating disorders children [22]. As addressed in the PARDI questionnaire, scales for affected sensory profiles were higher in individuals with a long period of symptoms, as well as an interest in food for the concern profile and ARFID severity scale. What makes a difference in food preferences in children is that while patients who are picky eaters ask for preferred types of meals, ARFID patients do not engage themselves in a relationship with food regardless of the type of nutrient offered during daytime [23]. Patients in similar studies exhibited an inverted relation between age and nutritional status compared to our findings. In our study, ARFID individuals had poorer weight statuses in a chronic evolution, but Keery et al. reported significantly acute weight loss [24]. Acute onset of symptoms often brings families and patients in for evaluation, and if proper management is offered, a nutritional balance might be restored sooner, while for a chronic evolution, BMI depreciation is less sharp. Nutritional support should begin as soon as a child is identified as being at risk in order to establish a balanced diet, and improve the patient’s eating habits and sensorial skills. Some clinical aspects of eating disorders are connected to gastrointestinal functioning through a good diversity of gut microbiota [25], and similar to functional gastrointestinal disorders in children, where there is an increased number of E. coli species [26], individuals with ARFID should be explored in terms of dysbiosis and therapeutic modulation. Since proteins are important modulators of growth and inflammation [27,28], there should be a special focus on dietary intake for children with ARFID in terms of early corrections of any deficits.

There were some limitations of the study due to modest sample sizes of ARFID patients, which prohibited us from further generalized conclusion on psychological differences between individuals. A future goal might include collaboration with a psychiatric hospital in order to determine the difference between children at risk of ARFID and a clear diagnosis due to so many clinical and behavioural manifestations. To conclude, this study has shown that there should be greater paediatric awareness in terms of feeding disorders screening and diagnostic tools, mainly in younger individuals and those with long-term symptom onset.

Some of individuals with restricted diets might experience other poorly symptomatic mechanisms, such as fructose intolerance, taking into account the association of several intolerances previously observed; food refusal can serve as a therapeutic tool for these patients [29]. Up to 94% of children with food allergies experience further feeding disorders as it is shown in some previous data [30]. In our study, nearly 90% had an allergy history, mainly in younger patients and in children with chronic intake issues, but it is difficult for practitioners to evaluate whether food refusal is a direct result of primary food restriction.

Although the diagnostic criteria for eating disorders such as ARFID are currently available, there is still a debate regarding aspects of weight loss and significant nutritional deficiencies as a consequences of food refusal when other medical or psychiatric disorders are eliminated. There is a high addressability for nutrition intervention in children with eating disorders in terms of acute and chronic therapy. Classically, intervention methods are represented by strategies to increase caloric volume with nutritional dense formula, tube feeding as a temporary measure and sometimes gastrostomy considerations, while hydration and growth is monitored regularly [31]. In our study, all kind of feeding strategies were used, in children considered to be at risk for ARFID, some of them required parenteral nutrition or gastrostomy. The ones needing more parenteral nutrition mainly had various acute exacerbations with a prolonged evolution, taking into account their nutritional status. Further intervention during dietary supplementation required tube withdrawal, and the typical approach was to increase the oral caloric intake while reducing the frequency of tube feeding. There is current lack of integrated management for these patients, although due to multifactorial aetiology, it usually requires a multidisciplinary team in order to reduce the time of nutrition therapy as much as possible [32]. The rehabilitation team should have a dietician with paediatric patient expertise and knowledge in terms feeding and eating disorders who aim to identify the most common mistakes of critically addressing the foods eaten by the patient and trying to change their eating habits. As mentioned before, in cases such as children patients with ARFID, the first step cannot be to change to the products consumed by the normal population, such as healthy food choices or a diet [33]. In the case of this type of eating disorder, when patients eat mainly unhealthy food, with a high amount of processed meals that are considered unsuitable for a normal diet, intervention cannot be conducted to dramatically withdraw patient food options, such as simply offering a newer and healthier alternative. If such an attitude is promoted, the outcome will eventually end in patients choosing extreme hunger. Management should initially include accepting unhealthy foods by the rehabilitation team and introducing newer energy dense formulas.

Although there should be a sustained strategy while treating ARFID, such as “family-based treatment” (FBT), which was described by the researcher team lead by Rosania and Lock [34], sometimes nutritional intervention in an invasive manner is necessary. FBT is widely used for other feeding disorders, such as anorexia and bulimia, and it refers to family implications conducted in three stages: first, there is a focus on restoring patient’s weight by parent implication in family meals, allowing the therapist to assess the family’s lifestyle; second, parents need to learn to control the child’s food reactions and should be empowered to manage the child’s symptoms; and third stage, when the patient’s weight is restored and they have already managed their eating behaviour, therapy is focused on regaining social and environmental reintegration [34].

## 5. Conclusions

Feeding disorders are an important issue among paediatricians, and a proper awareness for addressing these patients should be included in daily practice. To the best of our knowledge, this is the first evaluation of feeding disorders in the paediatric population in our country from a paediatric point of view, thus raising awareness for this issue in a special population. Therefore, it is likely that food avoidance, as it is currently conceptualised, is on a continuum and thus has dimensional rather than discrete diagnostic categories.

## Figures and Tables

**Figure 1 nutrients-15-02831-f001:**
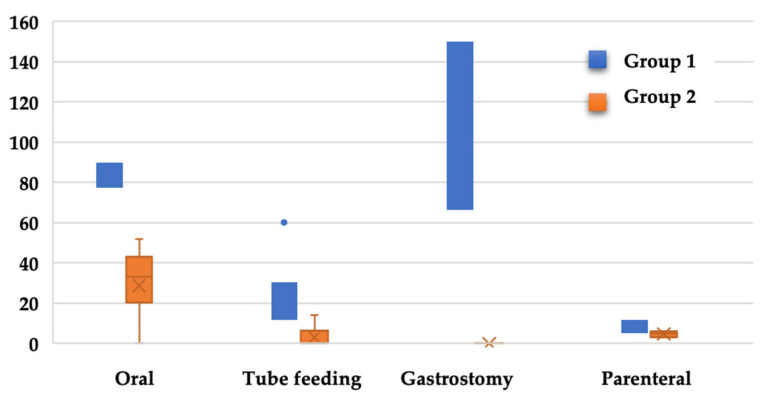
Duration of nutritional intervention according to group distribution.

**Table 1 nutrients-15-02831-t001:** Baseline characterization of the two study groups.

Characteristics	Group 1	Group 2	*p*-Value
Area (U/R), *n*	33/16	40/9	0.105
Sex (M/F), *n*	22/27	20/29	0.683
Age (years) *	6.91 ± 3.25	7.49 ± 2.91	0.448
Gestational period (normal/pathologic), *n*	37/12	41/8	0.316
Maternal age (years) *	29.43 ± 3.81	27.03 ± 3.04	<0.001
Father’s age (years) *	34.09 ± 5.77	31.88 ± 3.94	0.004
Mother’s level of education, *n* (%)			
None/Primary	8 (16.32)	6 (12.24)	0.383
Secondary/College	14 (28.57)	14 (28.57)	0.548
Higher	27 (55.1)	29 (59.18)	0.453
Living standard, *n* (%)			
Low	11 (22.44)	10 (20.4)	0.562
Medium	23 (46.93)	28 (57.14)	0.312
High	15 (30.61)	11 (22.44)	0.432
Marital status, *n* (%)			
Married/Cohabiting	13 (26.53)	25 (51.02)	0.001
Single/Divorced	36 (73.46)	24 (48.97)	0.001
Birth weight (g) *	3095.2 ± 478.77	3163.67 ± 360.83	0.213
BMI Score Z, *n* (%)			
−1 to +1	29 (59.18)	47 (95.91)	0.005
−1 to −2	16 (32.65)	2 (4.08)	0.001
Less than −2	6 (12.24)	0 (0)	0.001
Nutrition in the first 6 months (BF/FF/MF), *n*	18/25/6	22/14/13	0.005
Successful weaning, *n* (%)	28 (57.14)	45 (91.83)	0.001
Regular feeding, *n* (%)	20 (40.81)	33 (67.34)	0.005
Food allergy, *n* (%)	44 (89.79)	26 (53.06)	<0.001

*n*—number; *—values as means ± SD (standard deviation).

**Table 2 nutrients-15-02831-t002:** Age intervals and duration of symptoms distribution.

All N = 49	Age ^a^ < 5 Years N = 23 (46.93%)	Age ^a^ > 5 Years N = 25 (51.02%)	*p* ^b^	^c^ Symptom Onset < 12 Months N = 28 (57.14%)	^c^ Symptom Onset > 12 Months N = 21 (42.85%)	*p* ^b^
Demographics						
Age, years	3.6 ± 2.65	10.2 ± 2.60	<0.001	6.25 ± 3.65	4.24 ± 2.29	0.005
Gender, *n* (%)						
Male	10 (20.40)	11 (22.44)	0.567	9 (18.36)	13 (26.53)	0.005
Female	13 (26.53)	14 (28.57)	0.645	19 (38.77)	8 (16.32)	0.005
Allergy history, *n* (%)	33 (75)	11 (25)	<0.001	25 (56.81)	19 (43.18)	0.253
PARDI ^a^						
Sensory profile	6.2 ± 1.4	4.4 ± 1.3	<0.001	6.2 ± 3.1	7.4 ± 3.2	<0.001
Lack of interest profile	5.4 ± 3.4	4.6 ± 1.6	<0.001	5.4 ± 2.1	8.4 ± 3.5	<0.001
Concern profile	5.8 ± 2.3	4.4 ± 1.8	<0.001	5.4 ± 2.3	7.2 ± 3.5	<0.001
ARFID severity scale	6.4 ± 2.5	5.6 ± 2.4	0.05	4.2 ± 1.2	6.8 ± 2.6	<0.001
Nutritional status ^a^						
Kilograms	15.22 ± 1.16	18.16 ± 2.13	0.001	19.33 ± 2.66	16.55 ± 1.22	0.005
BMI	13.11 ± 1.44	16.36 ± 2.65	0.001	15.44 ± 1.22	14.33 ± 1.33	0.256
BMI Z score	−1.94 ± 0.33	−1.10 ± 0.88	0.05	−1.22 ± 0.23	−1.88 ± 0.78	0.05

^a^ Values are in means ± SD; ^b^ using Pearson’s chi-square test for categorical variables and *t*-test for continuous variables; ^c^ using Fischer exact test.

**Table 3 nutrients-15-02831-t003:** Distribution of variables related to eating patterns according to eating disorder risk.

	Group 1	Group 2	Statistics	*p*-Value
Number of meals, *n* (%)				
3	28 (57.14%)	6 (12.24%)	χ^2^ = 22.550	0.001
4	4 (8.16%)	16 (12.24%)		
5	2 (4.08%)	20 (40.81%)		
At least 3 meals per day				
Yes	35 (71.42%)	42 (85.71%)	χ^2^ = 2.618	0.077
No	14 (28.57%)	7 (14.28%)		
Breakfast				
Yes	41 (83.67%)	47 (95.91%)	χ^2^ = 2.127	0.462
No	8 (16.32%)	2 (4.08%)		
Special diet restriction				
Yes	35 (71.42%)	40 (81.63%)	χ^2^ = 38.376	0.001
No	14 (28.57%)	9 (18.36%)		
Exercise				
Yes	22 (44.89%)	19 (38.77%)	χ^2^ = 2.176	0.377
No	27 (55.10%)	30 (61.22%)		
Exercise frequency				
Everyday	7 (14.28%)	5 (10.20%)	χ^2^ = 2.544	0.314
1–2 times/week	11 (22.44%)	12 (24.48%)		
3 or more times/week	4 (8.16%)	2 (4.08%)		

## Data Availability

No new data were created or analysed in this study. Data sharing is not applicable to this article.

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
