# Peer review of "Risk Factors Related to Eating Disorders in a Romanian Children Population"

_nutrients, 2023, doi:10.3390/nu15132831_

Round 1

Reviewer 1 Report

This research, aimed at describing and comparing the characteristics of children with Avoidant/Restrictive Food Intake Disorder (ARFID) with healthy children, is important and intriguing. However, it is suggested that the description of the methodology is lacking in clarity and thoroughness, necessitating revisions.

Specifically, it is unclear whether this study was designed as a prospective study comparing ARFID children with healthy children. If this is a prospective study, the authors should provide more details on aspects such as whether they performed sample size calculations.

Furthermore, it appears that the ARFID children and healthy children were sampled at a 1:1 ratio, but the criteria for selecting the healthy children as controls are not explained. Information about the matching process (for instance, were the children seen on the same day or matched by age) is not provided. The selection criteria for the controls are also unclear from the present paper.

The results are similarly ambiguous. For example, the statement, "According to feeding disorders on admission, a number of 125 patients were initially selected, out of which 101 (80.8 %) meet the inclusion criteria," is confusing. It's not clear if the 125 refers to the number of individuals who met the inclusion criteria of "(1) aged 2-10 years; (2) presentation for evaluation of eating disorder symptoms," and if the 101 refers to those diagnosed with ARFID by pediatric gastroenterologists.

Given the vagueness of the methodology, it's difficult to thoroughly evaluate the overall paper. It is recommended that the authors provide more clarity in their methodology to make the results more understandable and reliable.

Author Response

Point 1: This research, aimed at describing and comparing the characteristics of children with Avoidant/Restrictive Food Intake Disorder (ARFID) with healthy children, is important and intriguing. However, it is suggested that the description of the methodology is lacking in clarity and thoroughness, necessitating revisions. Specifically, it is unclear whether this study was designed as a prospective study comparing ARFID children with healthy children. If this is a prospective study, the authors should provide more details on aspects such as whether they performed sample size calculations.

Response 1: This was a prospective, case control study comparing patients with food refusal to normal healthy controls, sampled 1:1, matched by age. It included children presented for feeding difficulties

Point 2: Furthermore, it appears that the ARFID children and healthy children were sampled at a 1:1 ratio, but the criteria for selecting the healthy children as controls are not explained. Information about the matching process (for instance, were the children seen on the same day or matched by age) is not provided. The selection criteria for the controls are also unclear from the present paper.

Response 2: This was a prospective, case control study comparing patients with food refusal to normal healthy controls, sampled 1:1, matched by age. It included children presented for feeding difficulties. Inclusion criteria for study population were: (1) aged 2-10 years; (2) presentation for evaluation of eating disorder symptoms (3) informed consent signed by the caregiver. In comparison, Group 2 included patients presented in the Ambulatory area of the same settling for regular paediatric visits, randomly selected, without any disturbances of food intake.

Point 3: The results are similarly ambiguous. For example, the statement, "According to feeding disorders on admission, a number of 125 patients were initially selected, out of which 101 (80.8 %) meet the inclusion criteria," is confusing. It's not clear if the 125 refers to the number of individuals who met the inclusion criteria of "(1) aged 2-10 years; (2) presentation for evaluation of eating disorder symptoms," and if the 101 refers to those diagnosed with ARFID by pediatric gastroenterologists.

Response 3: There was initially a number of 125 patients presented for feeding difficluties, out of which 101 (80.8 %) were admitted to the study population, after exclusion individual with medical chronic conditions.

Point 4: Given the vagueness of the methodology, it's difficult to thoroughly evaluate the overall paper. It is recommended that the authors provide more clarity in their methodology to make the results more understandable and reliable.

Response 4: This was a prospective, case control study comparing patients with food refusal to normal healthy controls, sampled 1:1, matched by age. It included children presented for feeding difficulties that were initially recruited during a two-year period (2020-2022), in the SARS-CoV-2 pandemic, from a large cohort of paediatric population addressed to the paediatric gastroenterology service in „Dr. Victor Gomoiu” Clinical Children Hospital from Bucharest, the capital of Romania. Inclusion criteria for study population were: (1) aged 2-10 years; (2) presentation for evaluation of eating disorder symptoms (3) informed consent signed by the caregiver. This population was considered Group 1, as individuals having risk for ARFID, identified by paediatric gastroenterologist. In comparison, Group 2 included patients presented in the Ambulatory area of the same settling for regular paediatric visits, randomly selected, without any disturbances of food intake. Patients with acute or chronic gastrointestinal disorders or with any other criteria for co-morbid medical disorders known to influence eating or weight were excluded from both groups. For secondary outcomes, patients at risk for ARFID were divided according to age, meaning less than 5 years and between 5-10 years. Moreover, according to duration of symptoms, the population included in the study was assessed in the two different groups, meaning duration of symptoms less than 12 months and over 12 months. Data collection was based on face-to face questionnaires or telephone interviews, performed by the same paediatric gastroenterologist. In the procedure, caregivers were assured of the voluntary nature of participation and confidentiality of the research and provided informed consent, before entering the study protocol. The research project was approved by the Hospital Committee of Ethics, no. 1758/01.02.2022, that reviewed the study design and all informational material.

Reviewer 2 Report

I looked at the Materials and Methods chapter. For me, the chapter "Participants and procedures" is quite unclear, many questions arise from it, which makes it pointless to read carefully the rest of the article.

It should be clearly written what kind of study it is. As it is written, it would be a case-control study.

If this is a case-control study, then I think it is an extremely poor choice of control group. The only inclusion criteria for control is the selection of the same number of controls as the number of cases, without any other parameters on the similarity of case and control. Or it was done, but not written, i.e. explained! The author has the right to choose any control group he wants, even the one described. But conclusions made on the basis of such a control group are not credible. If this is not a case-control study, then I ask the authors to write which study it is.

Likewise, it is not written how the subjects-cases from the pediatric cohort group were selected (random selection method and similar)

Was there a reason why a more detailed breakdown by age was not made as I feel these age groups are too large? This is my suggestion, but the authors decide on this.

Lines 23-24 and 59-62:

Is the conclusion in the abstract generalizable to the entire target pediatric population from which the case sample was drawn? I can guess. But if that's the case, then the sentence is questionable because I'm not sure the sample is representative. Due to the vaguely written design of the study, the aims of the study are also questionable if the selection of cases is not representative.

Author Response

Response to Reviewer 2 Comments

Point 1: I looked at the Materials and Methods chapter. For me, the chapter "Participants and procedures" is quite unclear, many questions arise from it, which makes it pointless to read carefully the rest of the article.It should be clearly written what kind of study it is. As it is written, it would be a case-control study. If this is a case-control study, then I think it is an extremely poor choice of control group.

Response 1: This was a prospective, case control study comparing patients with food refusal to normal healthy controls, sampled 1:1, matched by age. It included children presented for feeding difficulties that were initially recruited during a two-year period (2020-2022),

Point 2: The only inclusion criteria for control is the selection of the same number of controls as the number of cases, without any other parameters on the similarity of case and control. Or it was done, but not written, i.e. explained! The author has the right to choose any control group he wants, even the one described. But conclusions made on the basis of such a control group are not credible. If this is not a case-control study, then I ask the authors to write which study it is.

Response 2: This was a prospective, case control study comparing patients with food refusal to normal healthy controls, sampled 1:1, matched by age. It included children presented for feeding difficulties that were initially recruited during a two-year period (2020-2022), in the SARS-CoV-2 pandemic, from a large cohort of paediatric population addressed to the paediatric gastroenterology service in „Dr. Victor Gomoiu” Clinical Children Hospital from Bucharest, the capital of Romania. Inclusion criteria for study population were: (1) aged 2-10 years; (2) presentation for evaluation of eating disorder symptoms (3) informed consent signed by the caregiver. This population was considered Group 1, as individuals having risk for ARFID, identified by paediatric gastroenterologist. In comparison, Group 2 included patients presented in the Ambulatory area of the same settling for regular paediatric visits, randomly selected, without any disturbances of food intake. Patients with acute or chronic gastrointestinal disorders or with any other criteria for co-morbid medical disorders known to influence eating or weight were excluded from both groups. For secondary outcomes, patients at risk for ARFID were divided according to age, meaning less than 5 years and between 5-10 years. Moreover, according to duration of symptoms, the population included in the study was assessed in the two different groups, meaning duration of symptoms less than 12 months and over 12 months. Data collection was based on face-to face questionnaires or telephone interviews, performed by the same paediatric gastroenterologist. In the procedure, caregivers were assured of the voluntary nature of participation and confidentiality of the research and provided informed consent, before entering the study protocol. The research project was approved by the Hospital Committee of Ethics, no. 1758/01.02.2022, that reviewed the study design and all informational material.

Point 3: Likewise, it is not written how the subjects-cases from the pediatric cohort group were selected (random selection method and similar)

Response 3: In comparison, Group 2 included patients presented in the Ambulatory area of the same settling for regular paediatric visits, randomly selected, without any disturbances of food intake

Point 4: Was there a reason why a more detailed breakdown by age was not made as I feel these age groups are too large? This is my suggestion, but the authors decide on this.

Response 4: Age group were considered according to prescholar and prepubertal.

Point 5: Lines 23-24 and 59-62: Is the conclusion in the abstract generalizable to the entire target pediatric population from which the case sample was drawn? I can guess. But if that's the case, then the sentence is questionable because I'm not sure the sample is representative. Due to the vaguely written design of the study, the aims of the study are also questionable if the selection of cases is not representative.

Response 5: This article rise awarness among proper definition and rosk factors related to food intake disorders in the study population.
